# Resection Margin Status and Long-Term Outcomes after Pancreaticoduodenectomy for Ductal Adenocarcinoma: A Tertiary Referral Center Analysis

**DOI:** 10.3390/cancers16132347

**Published:** 2024-06-26

**Authors:** Giuseppe Quero, Davide De Sio, Claudio Fiorillo, Chiara Lucinato, Edoardo Panza, Beatrice Biffoni, Lodovica Langellotti, Vito Laterza, Giulia Scaglione, Flavia Taglioni, Giuseppe Massimiani, Roberta Menghi, Fausto Rosa, Teresa Mezza, Sergio Alfieri, Vincenzo Tondolo

**Affiliations:** 1Pancreatic Surgery Unit, Fondazione Policlinico Universitario “Agostino Gemelli” IRCCS, 00168 Rome, Italy; giuseppe.quero@policlinicogemelli.it (G.Q.); claudio.fiorillo@policlinicogemelli.it (C.F.); chiara.lucinato01@icatt.it (C.L.); edoardo.panza01@icatt.it (E.P.); beatrice.biffoni01@icatt.it (B.B.); lodovica.langellotti01@icatt.it (L.L.); vito.laterza01@icatt.it (V.L.); flavia.taglioni01@icatt.it (F.T.); giuseppe.massimiani01@icatt.it (G.M.); roberta.menghi@policlinicogemelli.it (R.M.); fausto.rosa@policlinicogemelli.it (F.R.); sergio.alfieri@policlinicogemelli.it (S.A.); 2Gemelli Pancreatic Center, CRMPG (Advanced Pancreatic Research Center), Fondazione Policlinico Universitario “Agostino Gemelli” IRCCS, 00168 Rome, Italy; teresa.mezza@policlinicogemelli.it; 3Dipartimento di Scienze Mediche e Chirurgiche, Università Cattolica del Sacro Cuore di Roma, 00168 Rome, Italy; vincenzo.tondolo@policlinicogemelli.it; 4Unità Operativa Complessa Anatomia Patologica Generale, Dipartimento di Scienze della Salute della Donna, del Bambino e di Sanità Pubblica, Fondazione Policlinico Universitario “Agostino Gemelli” IRCCS, 00168 Rome, Italy; giulia.scaglione@policlinicogemelli.it; 5General Surgery Unit, Fatebenefratelli Isola Tiberina—Gemelli Isola, 00186 Rome, Italy

**Keywords:** resection margin, pancreaticoduodenectomy, pancreatic ductal adenocarcinoma, long-term outcomes

## Abstract

**Simple Summary:**

This study investigates the impact of resection margin (R) status on overall survival (OS) and disease-free survival (DFS) in patients undergoing pancreaticoduodenectomy (PD) for pancreatic ductal adenocarcinoma (PDAC). A retrospective analysis of 167 PD cases from 2012 to 2023 revealed that 62.8% achieved negative margins (R0), while 37.1% had positive margins (R1). Patients with R1 status had significantly lower OS (23 vs. 36 months, *p* = 0.003) and DFS (10 vs. 18 months, *p* = 0.004) compared to R0 patients. Multivariate analysis identified R1 status and positive lymph nodes (N+) as independent factors adversely affecting both OS and DFS. Specifically, among patients with N+ disease, R1 status was associated with a notably decreased DFS (10 vs. 16 months, *p* = 0.05). The study concludes that achieving R0 status during PD is crucial for improved long-term outcomes, emphasizing the importance of radical surgery, especially in patients with lymph node involvement.

**Abstract:**

The influencing role of resection margin (R) status on long-term outcomes, namely overall (OS) and disease-free survival (DFS), after pancreaticoduodenectomy (PD) for pancreatic ductal adenocarcinoma (PDAC) is not still clear. The aim of this study is to evaluate the prognostic impact of R status after PD and to define tumor characteristics associated with a positive resection margin (R1). All PDs for PDAC performed between 2012 and 2023 were retrospectively enrolled. The effect of R status, patient clinico-demographic features, and tumor features on OS and DFS were assessed. One-hundred and sixty-seven patients who underwent PD for PDAC were included in the study. R0 was achieved in 105 cases (62.8%), while R1 was evidenced in 62 patients (37.1%). R1 was associated with a decreased OS (23 (13–38) months) as compared to R0 (36 (21–53) months) (*p* = 0.003). Similarly, DFS was shorter in R1 patients (10 (6–25) months) as compared to the R0 cohort (18 (9–70) months) (*p* = 0.004), with a consequent higher recurrence rate in cases of R1 (74.2% vs. 64.8% in the R0 group; *p* = 0.04). In the multivariate analysis, R1 and positive lymph nodes (N+) were the only independent influencing factors for OS (OR: 1.6; 95% CI: 1–2.5; *p* = 0.03 and OR: 1.7; 95% CI: 1–2.8; *p* = 0.04) and DFS (OR: 1.5; 95% CI: 1–2.1; *p* = 0.04 and OR: 1.8; 95% CI: 1.1–2.7; *p* = 0.009). Among 111 patients with N+ disease, R1 was associated with a significantly decreased DFS (10 (8–11) months) as compared to R0N+ patients (16 (11–21) months) (*p* = 0.05). In conclusion, the achievement of a negative resection margin is associated with survival benefits, particularly in cases of N1 disease. In addition, R0 was recognized as an independent prognostic feature for both OS and DFS. This further outlines the relevant role of radical surgery on long-term outcomes.

## 1. Introduction

Despite the recent advancement in the multimodal and surgical treatment of pancreatic ductal adenocarcinoma (PDAC), long-term outcomes still remain dismal, with a poor overall survival (OS) and disease-free survival (DFS) [1,2,3]. This is due to several factors, such as the presence of perineural and lymphovascular infiltration, tumor size, and metastatic lymph nodes, recognized as independent influencing features on survival outcomes after resection [4,5,6]. Furthermore, debates are still present in the literature on the influencing role of resection margin status (R) on long-term outcomes [3]. Some authors identified R status as an independent prognostic factor after pancreaticoduodenectomy (PD), reporting a 5-year OS of 26% in cases of tumor-free resection margins (R0) as compared to 8% in case of a microscopically positive margins (R1) [5,7,8]. Conversely, other authors did not recognize R status as independently affecting survival after resection [9,10,11,12]. Such a discrepancy may be explained by the nonconsensual definition of R status. Indeed, the Union for International Cancer Control (UICC) defines R1 as the presence of tumor cells on the resection margin (0 mm) [13], while the British Royal College of Pathologists defines R1 as the presence of tumor cells within 1 mm of the resection margin [14]. In addition, more recent reports demonstrated long-term advantages, namely OS and DFS, in cases of a resection margin clearance of at least 1.5–2 mm [15,16]. These contrasting definitions have inevitably resulted in a high variation in terms of positive resection margin rate, with reports in the literature between 17% and 85% after PD [17,18,19,20].

Apart from the contrasting definitions of R status, two additional hypotheses have been postulated to justify the discrepant oncological outcomes reported on the bases of the R value. Firstly, R1 is frequently concomitant to other prognostic features such as the presence of metastatic lymph nodes that may effectively have a more significant influence on OS and DFS than R status [21,22,23]. Secondly, it is likely that tumor recurrence may be more related to an aggressive tumor biology rather than the presence of tumor cells on the resection margin, as evidenced by locoregional recurrence (LRR) occurring in 10% to 25% of patients [9,24]. Based on these premises, the aim of this study is to evaluate the impact of R status on long-term outcomes, namely OS and DFS, and to define the clinico-demographic and oncological features that may be associated with R1 after PD for PDAC in a tertiary referral center.

## 2. Materials and Methods

All patients who underwent PD for a histologically proven diagnosis of PDAC at the Fondazione Policlinico Universitario “Agostino Gemelli” IRCCS of Rome between January 2012 and December 2023 were retrospectively included in the study. Patients with other histopathological diagnoses, including malignant intrapapillary mucinous neoplasms, were excluded from enrollment. Moreover, given the potential influencing role of neoadjuvant treatment on long-term outcomes, patients who underwent neoadjuvant treatment (NAT) were excluded from the analysis. All data were retrospectively retrieved from prospectively maintained databases, and follow-up data were collected until April 2024.

The collected data included age; sex; tumor location; adjuvant therapy; and histopathological features, namely tumor diameter and grading, number of harvested and metastatic lymph nodes, evaluation of lymphovascular and angio-invasion and perineural infiltration, pTNM stage according to the 8th edition of the AJCC/UICC system [13], and R status.

From 2018 onwards, all cases were preoperatively discussed in a multidisciplinary tumor board in order to assess tumor resectability [25]. Contraindications to tumor resection were the preoperative detection of distant metastases, tumor infiltration of the celiac trunk, common hepatic artery or superior mesenteric artery, or encasement of the superior mesenteric vein/portal vein. Post-operatively, all cases were re-discussed at the multidisciplinary tumor board to determine the potential indication for adjuvant therapy. Since a consistent number of patients underwent adjuvant therapy at a site other than our institution, information on the type of adjuvant treatment (chemotherapy and/or radiotherapy), dose, and frequency was not available. Thus, the study included only if adjuvant therapy was recommended.

### 2.1. Surgical Technique

Details regarding the surgical procedure were previously reported [26,27]. All patients underwent a Whipple procedure with standard lymphadenectomy [28]. The division of the pancreatic head was followed by dissection of the retroportal lamina from the portal/superior mesenteric vein and along the right/anterior aspect of the superior mesenteric artery. The retroportal lamina margin was always macroscopically assessed and inked with methylene blue along its length for an appropriate pathological assessment.

### 2.2. Post-Operative Follow-Up and Long-Term Outcomes Analysis

Follow-up information was obtained from the electronical medical records of our institution, primary care physician, or through a direct telephonic contact with the patient or relatives. When performed at our institution, post-operative follow-up included physical examination, laboratory tests (including measurement of carbohydrate antigen 19.9 and carcinoembryonic antigen levels), and transabdominal ultrasound every 3 months for the first two years after PD. Computed tomography (CT) was prescribed every 6 months after surgery. A whole-body positron emission tomography (PET) scan with 18-fluoro-2-desoxy-glucose (FDG) was indicated in cases of an inconclusive diagnosis at the CT scan.

The evaluation of long-term outcomes included tumor recurrence, OS, and DFS. Local recurrence (LR) was defined as tumor relapse in the retroperitoneum at the pancreatic remnant, regional lymph nodes, and around the mesenteric vessels and/or celiac trunk, while tumor recurrence at any other site, such as the liver, lung, para-aortic lymph nodes, and peritoneal cavity, were defined as distant metastases. OS was defined as the time from surgery to the last follow-up, while DFS was defined as the time between surgery and the detection of tumor recurrence or death.

### 2.3. Pathological Assessment

Macroscopic and microscopic evaluation of the surgical specimens as well as the definition of the resection margins were conducted according to the recommendation of Verbeke et al. [17,29]. Specifically, resection margins identified and evaluated were the anterior and posterior pancreatic surfaces, medial (defined as the superior mesenteric vein root), retroportal lamina (the closest margin to the superior mesenteric artery), common bile duct, and pancreatic neck. R status was defined according to the Royal College of Pathologists [14]. Specifically, the microscopic resection margin was considered to be positive (R1) when tumor cells were detected within 1 mm of the transection margin. All the histopathological slides were retrospectively reviewed by a dedicated pathologist in order to reassess the R status [14] and reclassify the tumor staging according to the 8th edition of the AJCC/UICC TNM system 2018.

### 2.4. Study Outcomes

The primary aim of the study was to evaluate the impact of R status on the long-term outcomes after PD in terms of local recurrence, OS, and DFS. The secondary aim was a further definition of the clinico-demographic and oncological features related to R1 and their impact on long-term survival and recurrence.

### 2.5. Statistical Analysis

Continuous variables were reported as medians and quartile ranks (QRs) and categorical variables as numbers and percentages. Student’s *t* test, Mann–Whitney U test, Fisher’s test, and the χ^2^ test were used for the univariate analysis. A *p* value ≤ 0.05 was considered statistically significant. OS and DFS were calculated using the Kaplan–Meier method, and the log-rank test was employed for the evaluation of differences in recurrence and survival between groups. Features significantly related to recurrence or survival were included in a Cox proportional hazards regression analysis. Results were expressed as odds ratios (ORs) and 95% confidence intervals (CIs). Statistical significance was reached for a two-tailed *p* value < 0.05. SPSS version 25 for Windows (SPSS Inc., Chicago, IL, USA) was used to perform all tests.

## 3. Results

Between January 2012 and December 2023, 221 patients underwent PD for a histologically proven diagnosis of PDAC at the Pancreatic Surgery Unit of the Fondazione Policlinico Universitario Agostino Gemelli IRCCS of Rome. Of these, 39 (17.6%) underwent NAT, 6 (2.7%) died during hospitalization, and 9 (4%) were lost to follow-up and were excluded from the analysis. As a whole, 167 (75.6%) patients constituted the study cohort (Figure 1), with a nearly equal distribution between males (78–46.7%) and females (89–53.3%). The median age was 69 (60.5–75) years. The majority of the lesions were located in the pancreatic head (146–87.4%), 17 (10.2%) in the uncinate process and 4 (2.4%) in the isthmus. Margin negativity (R0 group) was reached in 105 patients (62.8%), while 62 patients (37.1%) presented at least one positive margin and, thus, constituted the R1 group. Table 1 reports the clinico-demographic characteristics of the study cohort.

### 3.1. Comparison between R0 and R1 Patients (Table 1)

Clinico-demographic features were comparable between the R0 and R1 cohorts. Specifically, no difference was documented in terms of tumor location (*p* = 0.48), grading (*p* = 0.32), and dimension (*p* = 0.54), while a more advanced T staging was evidenced in the R1 cohort (*p* = 0.004). A tangential venous resection was needed in 9 cases (5.4%) for an intraoperative suspicion of vascular invasion, with no difference between the R0 and R1 groups (*p* = 0.2). The median number of harvested lymph nodes was comparable between the two groups (*p* = 0.28). However, the R1 cohort more frequently presented metastatic lymph nodes (48–77.4%) as compared to R0 patients (63–60%) (*p* = 0.02). Moreover, perineural infiltration was more frequently encountered in cases of R1 resection (60–96.8% vs. 92–87.6% in cases of R0; *p* = 0.05). Similarly, the R1 group presented a higher rate of angio/lymphovascular invasion than the R0 cohort (*p* = 0.03). Adjuvant therapy was administered equally to R0 and R1 patients (*p* = 0.24). In this last regard, 6 R1 patients (9.7%) did not undergo adjuvant treatment due to patient refusal (4 cases) and poor post-operative performance status (2 cases). The site of margin positivity is reported in Appendix A.

### 3.2. Impact of R Status on Long-Term Outcomes

As a whole, tumor recurrence was evidenced in 114 (68.3%) patients after a median time of 10 (6–18) months: 21 (12.6%) had a local recurrence, 61 (36.5%) presented distant metastases, and 32 (19.2%) had local and distant recurrence. Notably, the R1 cohort presented a higher recurrence rate (74.2%–46 patients) as compared to the R0 cohort (64.8%–68 patients) (*p* = 0.04) with no difference in terms of median recurrence time between the R1 (9 (5–13) months)) and R0 groups (10 (6–18) months) (*p* = 0.18). No difference was noted between the two cohorts in terms of local, distant, and local and distant recurrence rate (*p* = 0.15), although a slightly higher rate of local recurrence was noted in the R1 population. Data on tumor relapse are reported in Table 2.

The median OS of the whole study population was 30 (17–46) months with a significantly longer survival in cases of R0 (36 (21–53) months) in comparison to R1 (23 (13–38) months) (*p* = 0.003). Similarly, the overall median DFS was 14 (7–33) months, with a significantly worse outcome in cases of R1 (10 (6–25) months) as compared to R0 (18 (9–70) months) (*p* = 0.004) (Figure 2A,B).

For OS, in the univariate analysis, R1 status (OR: 1.8; 95% CI: 1.1–2.7; *p* = 0.009), T staging (OR: 2.1; 95% CI: 1.2–3.8; *p* = 0.01), and N+ (OR: 2; 95% CI: 1.3–3.3; *p* = 0.004) resulted in negative prognostic factors. The multivariate analysis confirmed R1 status (OR: 1.6; 95% CI: 1–2.5; *p* = 0.03) and N+ (OR: 1.7; 95% CI: 1–2.8; *p* = 0.04) as factors independently associated with a worse OS (Table 3).

In the univariate analysis, R1 (OR: 1.7; 95% CI: 1.2–2.4; *p* = 0.006), T staging (OR: 1.7; 95% CI: 1–3.1; *p* = 0.02), and N+ (OR: 2; 95% CI: 1.3–3; *p* = 0.001) resulted in negative prognostic factors for DFS. As for OS, in the multivariate analysis, only R1 (OR: 1.5; 95% CI: 1–2.1; *p* = 0.04) and N+ status (OR: 1.8; 95% CI: 1.1–2.7; *p* = 0.009) were confirmed as factors independently affecting DFS (Table 3).

### 3.3. Correlation Analysis of N and R Status with Oncological Outcomes

A further analysis was conducted to assess the impact of margin and nodal status, combined or alone, on long-term outcomes, namely OS and DFS. Patients were grouped according to R status into lymph node negative (R0N0, 42 patients—25.1% and R1N0 14 patients −8.4%) and node positive (R0N+, 61 patients −36.5% and R1N+, 50 patients—29.9%).

Tumor characteristics according to the R and N status are reported in Table 4.

The R0N0 cohort presented a longer OS (37 (19–54) months) as compared to the R0N+ (27 (23–30) months; *p* = 0.005), R1N0 (34 (16–51) months; *p* = 0.01), and R1N+ (23 (17–28) months; *p*< 0.0001) groups (Figure 3A). Similarly, the R0N0 group had a significantly longer DFS (24 (16–32) months) as compared to the R0N+ (16 (11–21) months; *p* = 0.008) and R1N+ (10 (10–17) months; *p* < 0.0001) groups, while no difference was noted in comparison to the R1N0 cohort (20 (10–37) months; *p* = 0.27). Interestingly, R0N+ patients presented a longer DFS than R1N+ (16 months, (11–21) vs. 10 (8–11) months, respectively; *p* = 0.05) (Figure 3B).

## 4. Discussion

This study shows that a margin clearance of ≤1 mm significantly affects long-term survival after PD for PDAC as compared to a negative resection margin. Moreover, R1 was significantly related to a higher rate of tumor recurrence as compared to R0 patients, although no difference was noted in terms of median time of disease relapse. These findings give further support to the “1 mm rule”, according to which oncological radicality in PD for PDAC is guaranteed when at least 1 mm of clearance margin is reached. Although this recommendation is in line with several other reports in the literature [30,31,32], other authors did not recognize R status as an independent predictor of a poor prognosis [9,10,11,12]. These contrasting data may be explained by the different R status definitions currently used in the literature [13,14,15,16] and, above all, in the frequent concomitant presence of other confounding prognostic factors associated with R1, such as the tumor size, tumor location, and the presence of positive lymph nodes [21,22,23]. According to this last hypothesis, the concomitance of these other validated prognostic features would make the R status a consequence of a more advanced disease rather than a surrogate marker of tumor biology [17,33,34]. It is, thus, undeniable that the effect of positive resection margins on PDAC outcomes still remains controversial. In order to give our contribution to this ongoing debate, we performed a detailed analysis of tumor characteristics and outcomes according to the R status, along with a subcategorization of patients according to the concomitant presence of further features significantly associated, at the multivariate analysis, with a worse OS and DFS, namely N status. This additional analysis was aimed at effectively demonstrating the independent impact of margin and N status on both survival and recurrence.

Reviewing our data, R1 patients effectively presented a more advanced disease, with a higher tumor stage, a more frequent detection of metastastic lymph nodes, and perineural and angio/lymphovascular infiltration, probably reflecting a more aggressive tumor biology. These results are similar to those reported by Tummers et al. [7] and Kimbrough et al. [35], making the R1 status representative of more aggressive disease. As a consequence, our R1 population presented a poor overall survival of 23 (13–38) months as compared to 36 (21–53) months in cases of R0 (*p* = 0.003). However, independent of the coexistence of additional prognostic factors, R1 itself was an independent risk factor for a worse OS in the multivariate analysis (OR: 1.6; 95% CI: 1–2.5; *p* = 0.03), supporting the independent influencing role of R status on long-term outcomes. This inevitably underlines the key role of curative resection (R0) on OS after PD for PDAC [36,37,38] and makes patient selection for resection an essential step to guarantee an adequate long-term outcome. In this context, the benefits of systemic therapy for pancreatic tumors in terms of OS and DFS are highly recognized. Indeed, the impressive survival advantages of FOLFIRINOX chemotherapy are increasingly emphasized, and the clear benefits in terms of tumor downstaging and improvement in resection rates in borderline lesions are currently paving the way for its application to even resectable PDACs, although contrasting results are currently present in the literature on this last topic [39,40,41].

With regards to the correlation between R1 and disease recurrence, our analysis outlined a higher rate of tumor relapse (more than 70%) and a consequent poorer DFS (10 (6–25) months) in R1 patients as compared to the R0 cohort. Notably, although not statistically significant (probably due to the small sample size of the study cohorts), R1 was related to a higher rate of local recurrence (12 cases (19.4%) vs. 9 cases (8.6%) in the R0 population). Although these data are in line with the majority of evidence [7,35], some authors [9,42,43] did not recognize the resection margin status as affecting disease relapse. The postulation that a positive margin may be predictive of a higher risk of tumor recurrence makes sense, since the microscopic tumor residual may trigger a disease relapse. However, recent reports outlined a sort of selective correlation between the type of margin involved and a higher risk of tumor recurrence. Specifically, the “vascular” margin seems to be the most relevant [17,33,44,45] in comparison to the posterior margin and anterior surface involvement [34]. Similarly, Pingpank et al. [44], documented a significantly lower DFS in patients with vascular margin positivity than those with posterior or pancreatic margin involvement. However, the limited sample size of our study population did not permit a further subanalysis according to the type of positive margin encountered, although the majority of our R1 patients presented vascular margin positivity (superior mesenteric vein margin).

Lymph node involvement and the number of positive lymph nodes are well-known prognostic factors after PD [21,23] related to a worse OS and DFS. These findings are further confirmed in our cohort of analysis, according to which the detection of positive lymph nodes was directly correlated with a lower OS and a higher recurrence rate. More interestingly, the prognostic role of N+ was additionally evaluated in relation to the R status, evidencing worse long-term outcomes independent of the resection margin positivity or negativity. Similarly, a further confirmation of the independent prognostic role of R1 was documented in the comparison of DFS between R0N+ and R1N+ patients, highlighting a worse long-term outcome in cases of R1 and positive lymph nodes as compared to R0 and metastatic lymph nodes.

This study presents several limitations. Firstly, the retrospective study design may have introduced a selection bias in the study population. Secondly, for the analysis of margin clearance, the resection margins were grouped together. This significantly limited the specific evaluation of the influencing role of the different margins that may differently affect long-term outcomes, as evidenced by previous studies [33,34,46]. Furthermore, the limited sample size of the study population significantly reduced the number of patients grouped according to the R and N status, making the statistical comparison not fully reliable. Finally, data on adjuvant treatment were not available. This inevitably limited the evaluation of the impact of type and duration of the post-operative treatment on long-term outcomes.

## 5. Conclusions

Despite the above-mentioned limitations, this work substantially contributes to the ongoing debate on the prognostic role of R status after PD. Our analysis further supports survival benefits for patients with a margin clearance of at least 1 mm, independent of the concomitance of additional prognostic features. This inevitably highlights the relevant role that radical surgery has on long-term outcomes, making a tumor-free resection essential for patients. Nevertheless, there is an undeniable need for additional studies conducted in a prospective manner and according to predefined protocols for the definition of R status and for a uniform pathological analysis to further and definitively support our findings.

## Figures and Tables

**Figure 1 cancers-16-02347-f001:**
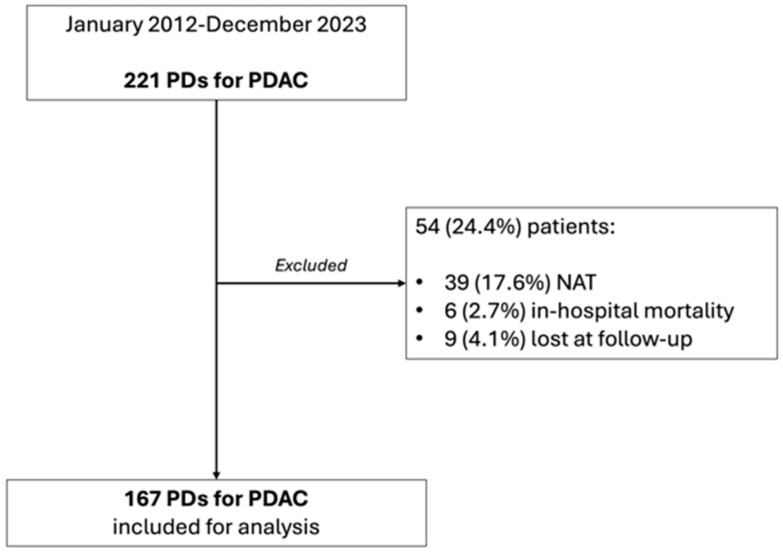
Study population flowchart.

**Figure 2 cancers-16-02347-f002:**
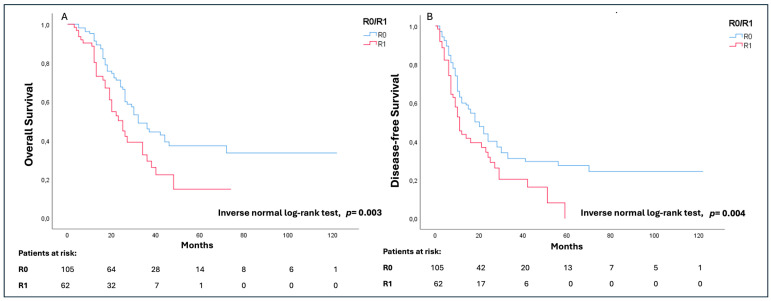
Overall survival (OS) (**A**) and disease-free survival (DFS) (**B**) analysis according to the R status.

**Figure 3 cancers-16-02347-f003:**
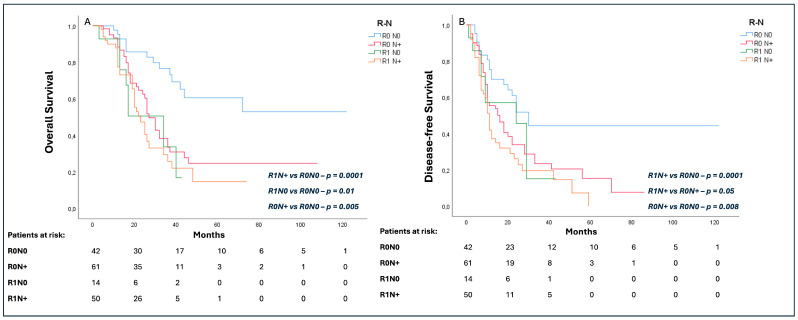
Subgroup analysis of effect of R and N status on overall survival (OS) (**A**) and disease-free survival (DFS) (**B**).

**Table 1 cancers-16-02347-t001:** Clinico-demographic characteristics of the study cohort and according to resection margin status.

Variables	Study Population (*n* = 167)	R0 (*n* = 105)	R1 (*n* = 62)	*p*
**Sex, *n (%)***				
*Male*	78 (46.7)	51 (48.6)	27 (43.5)	0.53
*Female*	89 (53.3)	54 (51.4)	35 (56.4)	
**Age (years), *median (QR)***	69 (60.5–75)	69 (60.5–75.0)	68 (59.5–73)	0.15
**Tumor location, *n (%)***				
*Head*	146 (87.4)	89 (84.8)	57 (91.9)	0.48
*Isthmus*	4 (2.4)	3 (2.9)	1 (1.6)	
*Uncinate process*	17 (10.2)	13 (12.4)	4 (6.4)	
**Major complications, *n (%)***	35 (21)	23 (21.9)	12 (19%)	0.36
**Tumor grading, *n (%)***				
*G1*	3 (1.8)	3 (2.9)	0 (0)	0.32
*G2*	129 (77.2)	81 (77.1)	48 (77.4)	
*G3*	35 (21)	21 (20)	14 (22.6)	
**Tumor dimension (mm), *median (QR)***	30 (23–35)	30 (22–35)	28 (25–35.2)	0.54
**T, *n (%)***				
*T1a*	1 (0.6)	1 (0.9)	0 (0)	**0.004**
*T1b*	0 (0)	0 (0)	0 (0)	
*T1c*	25 (15)	18 (17.1)	7 (11.3)	
*T2*	124 (74.2)	79 (75.2)	45 (72.6)	
*T3*	17 (10.2)	7 (6.7)	10 (16.1)	
*T4*	0 (0)	0 (0)	0 (0)	
**N, *n (%)***				
*N0*	56 (33.5)	42 (40)	14 (22.6)	**0.02**
*N+*	111 (66.5)	63 (60)	48 (77.4)	
**Vascular resection, *n (%)***	9 (5.4)	6 (5.7)	3 (4.8)	0.2
**Harvested lymph nodes, *median (QR)***	20 (16–28)	19 (16–26.5)	23 (15.7–28)	0.28
**Positive lymph nodes, *median (QR)***	2 (0–4)	1 (0–4)	2 (1–6)	**0.004**
**Perineural invasion, *n (%)***	152 (91)	92 (87.6)	60 (96.8)	**0.05**
**Angio/lympho-vascular invasion, *n (%)***	135 (80.8)	81 (77.1)	54 (87.1)	**0.03**
**Adjuvant therapy, *n (%)***	144 (86.2)	88 (83.8)	56 (90.3)	0.24

**Table 2 cancers-16-02347-t002:** Patterns of recurrence.

Patterns of Recurrence	Study Population (*n* = 167)	R0 (*n* = 105)	R1 (*n* = 62)	*p*
Recurrence, *n (%)*	114 (68.3)	68 (64.8)	46 (74.2)	0.04
Time to recurrence (months), *median (QR)*	10 (6–18)	10 (6–18)	9 (5–13)	0.18
Recurrence location, *n (%)*				0.15
*Local*	21 (12.6)	9 (8.6)	12 (19.4)	
*Distant*	61 (36.5)	37 (35.2)	24 (38.7)	
*Local + distant*	32 (19.2)	22 (21)	10 (16.1)	

**Table 3 cancers-16-02347-t003:** Univariate and multivariate analysis for OS and DFS.

Variable	OS				DFS			
	Univariate Analysis		Multivariate Analysis		Univariate Analysis		Multivariate Analysis	
	OR (95% CI)	*p*	OR (95% CI)	*p*	OR (95% CI)	*p*	OR (95% CI)	*p*
Sex	0.93 (0.61–1.41)	0.73			0.78 (0.54–1.14)	0.2		
Age > 65 years	1 (0.6–1.5)	0.92			1 (0.7–1.4)	0.95		
Major complications	1.2 (0.81–1.7)	0.12			1.6 (0.92–3.5)	0.07		
Vascular resection	0.81 (0.52–1.34)	0.64			0.67 (0.41–1.01)	0.4		
R1 status	1.8 (1.1–2.7)	0.009	1.6 (1–2.5)	0.03	1.7 (1.2–2.4)	0.006	1.5 (1–2.1)	0.04
Multiple R1	0.7 (0.3–2.0)	0.56			1.1 (0.5–2.4)	0.87		
T3 vs. T1–2	2.1 (1.2–3.8)	0.01	1.6 (0.9–3)	0.1	1.7 (1–3.1)	0.02	1.5 (0.9–2.6)	0.13
N+	2 (1.3–3.3)	0.004	1.7 (1–2.8)	0.04	2 (1.3–3)	0.001	1.8 (1.1–2.7)	0.009
Perineural invasion	1.5 (0.7–3.2)	0.24			1.5 (0.8–3)	0.19		
Angio/Lymphovascular invasion	1.2 (0.7–2.1)	0.46			1.2 (0.7–2)	0.77		
Adjuvant therapy	1 (0.5–1.9)	0.90			1.3 (0.7–2.3)	0.42		

**Table 4 cancers-16-02347-t004:** Histopathological characteristics according to R and N status.

Variables	R0N0 (*n* = 42)	R0N+ (*n* = 61)	R1N0 (*n* = 14)	R1N+ (*n* = 50)	*p*
Perineural infiltration, *n (%)*	32 (76.2)	58 (95.1)	14 (100)	48 (96)	0.001
Angio/lymphovascular invasion, *n (%)*	21 (50)	58 (95.1)	12 (85.7)	44 (88)	0.0001
G, *n (%)*					0.09
*1*	3 (7.1)	0	0	0	
*2*	33 (78.6)	46 (75.4)	12 (85.7)	38 (76)	
*3*	6 (14.3)	15 (24.6)	2 (14.3)	12 (24)	
Tumor dimension (mm), *median (IQR)*	25 (20–35)	31 (30–25)	23 (18–25)	30 (25–35)	0.01

## Data Availability

Data are available upon reasonable request at davide.desio01@gmail.com.

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
