# Peer review of "Resection Margin Status and Long-Term Outcomes after Pancreaticoduodenectomy for Ductal Adenocarcinoma: A Tertiary Referral Center Analysis"

_cancers, 2024, doi:10.3390/cancers16132347_

Round 1
Reviewer 1 Report
Comments and Suggestions for Authors
(1) No innovations/new techniques are offered and the findings and conclusions were expectable.
(2) The presented evaluation has the drawbacks of a retrospective study design.
(3) Please check the numbers: In the "Results" section (line 201), you write that R0 patients had a median OS of 36 (21-53) months; but in the discussion (line 266) you write that they had a median OS of 30 (17-46) months.
(4) Please check the references/citations: In the discussion (line 291) "Pinkpank et al." is cited as [48], but in the reference list it has number 47.
(5) An evaluation of the R1 subgroups would be interesting, but the sample size is too small.
(6) As neoadjuvant therapy is on the rise, an evaluation of these patients would also be interesting, but, again, the sample is small.
(7) No precise information on adjuvant treatment is provided.
(8) Larger/multicenter studies would allow more reliable and detailed evaluations.
Additional Comments/Suggestions:
(9) Introduction, line 75: "R1is..." -> R1 is...
(10) Statistical analysis, line 155: "the log rank test was used the evaluation of differences in recurrence and survival between groups" -> the log rank test was used for the evaluation of differences in recurrence and survival between groups.
(11) Statistical analysis, line 157: "regression ana-lysis" -> regression analysis.
(12) Comparison between R0 and R1 patients, line 183: "The site of margin positivity is reported a Supplementary Table 1" -> The site of margin positivity is reported in Supplementary Table 1.
(13) Impact of R status on long-term outcomes, line 215: "As for OS, at the multivariate analysis, only R1 (…) and N+ (…) as factors independently affecting DFS (Table 3)" – this statement is incomplete.
(14) Correlation analysis of N and R status with oncological outcomes, line 229: ")" is missing after "30".
(15) Discussion, line 241: "despite" -> although.
(16) Discussion, line 269: "This inevitably underline the key role..." > This inevitably underlines the key role...
(17) Discussion, line 271: "patients selection" -> patient selection.
(18) Discussion, line 273: "is" -> are.
(19) Discussion, line 274: "is increasingly" -> are increasingly.
(20) Discussion, line 275: "is currently" -> are currently.
(21) Discussion, line 283: "12 cases -19.4% vs 9 cases (8.6%)" -> 12 cases (19.4%) vs 9 cases (8.6%).
(22) Discussion, line 284: "evidences" -> evidence.
(23) Discussion, line 297: "lymph nodes involvement" -> lymph node involvement.
(24) Discussion, line 303: "a further confirm" -> a further confirmation.
(25) Discussion, line 304: "R1was..." -> R1 was...
Comments on the Quality of English Languagegenerally well readable
Author Response
GENERAL CONSIDERATIONSPrimarily we would like to thank the Editor and the Reviewers for taking the time to thoughtfully consider our manuscript, and for providing precious observations and suggestions, which have been very helpful to amend the manuscript, as we hope it emerges from the examination of the present document.
It is our feeling that along the line of reviewing the paper we also managed to improve it both in presentation and readability. Please find below our point-by-point response.
Reviewer #1
Comments to the Authors
Comment #1.1: No innovations/new techniques are offered and the findings and conclusions were expectable.
Answer #1.1: We do thank the reviewer for this comment, and we partly agree with this observation. Indeed, although the topic may be considered no innovative, it is important to underline that no conclusive results are present in the literature on the prognostic role of the R status. Specifically, even comparing studies that used the same definition of R status (such as ref. 5-8 and 12), the impact of R1 on long-term outcomes significantly varied and is, thus, not yet clear. This outlines the need for additional investigations on this topic especially derived from tertiary referral centers like ours. For these reasons (reported in the Introduction section) we decided to give our contribution.
Comment #1.2: The presented evaluation has the drawbacks of a retrospective study design.
Answer #1.2: We thank the reviewer for this comment, and we do agree that the retrospective study design may represent a potential bias. In this regard, we already highlighted such a limitation in the Discussion section at lines 315-316 as follows:
“This study presents several limitations. Firstly, the retrospective study design may have introduced a selection bias in the study population.”
Comment #1.3: Please check the numbers: In the "Results" section (line 201), you write that R0 patients had a median OS of 36 (21-53) months; but in the discussion (line 266) you write that they had a median OS of 30 (17-46) months.
Response #1.3: We do apologize for the typo. The correct OS value for R0 is 36 (21-53) months. Please find the correct OS value in the Discussion section (new line 274), as follows:
“36 (21-53) months in case of R0 (p=0.003).”
Comment #1.4: Please check the references/citations: In the discussion (line 291) "Pinkpank et al." is cited as [48], but in the reference list it has number 47.
Response #1.4: We do apologize for the mistake. The correct reference for “Pinkpank et al.” has been added (new line 299 of the Discussion section).
Comment #1.5: An evaluation of the R1 subgroups would be interesting, but the sample size is too small.
Response #1.5: We do agree on this observation (as we reported in the Discussion section at lines 316-319), especially in order to better define the impact of the different margin positivity on long-term outcomes. However, as the reviewer already underlined, the sample size of the R1 group is too small to permit to draw solid conclusions. However, our objective, in the near future, is to conduct a multicenter study to gain a sufficient sample size in order to investigate the prognostic role of the different margins.
Comment #1.6: As neoadjuvant therapy is on the rise, an evaluation of these patients would also be interesting, but, again, the sample is small.
Response #1.6: We thank the reviewer for this comment, and we totally agree with this observation.
Indeed, the indication to NAT for pancreatic cancer is progressively increasing over the years and new protocols are ongoing even for resectable tumors. However, two main drawbacks induced us to exclude patients who underwent NAT from our study cohort. Firstly, as reported by the reviewer, the sample size is small to obtain a valuable evaluation. Second, during the study period we analyzed (2012-2023), NAT significantly evolved and different NAT schemes were used over the years. This would imply a significant bias especially in consideration to the long-term outcomes evaluation.
However, as reported in the previous comment, we are designing a multicenter study protocol that will include NAT as an additional feature to analyze.
Comment #1.7: No precise information on adjuvant treatment is provided.
Response #1.7: We do thank the reviewer for this observation that permits us to further clarify such a limitation. As reported in lines 104-108 of the manuscript, a consistent number of patients underwent adjuvant treatment at a site other than our institution, significantly limiting the information on the type, dose and frequency of the post-operative treatment. It is, however, important to further outline this drawback of the study. Thus, the following statement was added to the Discussion section at pag. 9 (lines 322-324):
“Finally, data on adjuvant treatment were not available. This inevitably limited the evaluation of the impact of type and duration of the post-operative treatment on long-term outcomes.”
Comment #1.8: Larger/multicenter studies would allow more reliable and detailed evaluations.
Response #1.8: We do agree with this comment and, as reported in previous comments, we already working on a multicenter protocol in order to increase the sample size. This would permit us to analyze the potential impact of further features, such as NAT and type of margin, on long-term outcomes.
Comment #1.9: Introduction, line 75: "R1is..." -> R1 is...
Response #1.9: We added the missing space.
Comment #1.10: Statistical analysis, line 155: "the log rank test was used the evaluation of differences in recurrence and survival between groups" -> the log rank test was used for the evaluation of differences in recurrence and survival between groups.
Response #1.10: We corrected the sentence at the new line 154.
Comment #1.11: Statistical analysis, line 157: "regression ana-lysis" -> regression analysis.
Response #1.11: We corrected the word “analysis”.
Comment #1.12: Comparison between R0 and R1 patients, line 183: "The site of margin positivity is reported a Supplementary Table 1" -> The site of margin positivity is reported in Supplementary Table 1.
Response #1.12: We corrected the sentence at the new line 194.
Comment #1.13: Impact of R status on long-term outcomes, line 215: "As for OS, at the multivariate analysis, only R1 (…) and N+ (…) as factors independently affecting DFS (Table 3)" – this statement is incomplete.
Response #1.13: We apologize for the forgetfulness. We completed the sentence at the new line 225 by adding the missing word:
“As for OS, at the multivariate analysis, only R1 (OR: 1.5; 95% CI: 1-2.1; p = 0.04) and N+ status (OR: 1.8; 95% CI: 1.1-2.7; p = 0.009) were confirmed as factors independently affecting DFS (Table 3).”
Comment #1.14: Correlation analysis of N and R status with oncological outcomes, line 229: ")" is missing after "30".
Response #1.14: We added the missing digit.
Comment #1.15: Discussion, line 241: "despite" -> although.
Response #1.15: We apologize for the mistake; we corrected the sentence at the new line 249-250, as suggested.
Comment #1.16: Discussion, line 269: "This inevitably underline the key role..." > This inevitably underlines the key role...
Response #1.16: We added the missing digit.
Comment #1.17: Discussion, line 271: "patients selection" -> patient selection.
Response #1.17: We removed the “s”
Comment #1.18: Discussion, line 273: "is" -> are.
Response #1.18: We corrected the sentence at the new line 281 as suggested
Comment #1.19: Discussion, line 274: "is increasingly" -> are increasingly.
Response #1.19: We corrected the sentence at the new line 282 as suggested
Comment #1.20: Discussion, line 275: "is currently" -> are currently.
Response #1.20: We corrected the sentence at the new line 283 as suggested
Comment #1.21: Discussion, line 283: "12 cases -19.4% vs 9 cases (8.6%)" -> 12 cases (19.4%) vs 9 cases (8.6%).
Response #1.21: We corrected the new line 291 by adding the missing digits.
Comment #1.22: Discussion, line 284: "evidences" -> evidence.
Response #1.22: We changed the plural form with the singular one.
Comment #1.23: Discussion, line 297: "lymph nodes involvement" -> lymph node involvement.
Response #1.23: We changed the plural form with the singular one.
Comment #1.24: Discussion, line 303: "a further confirm" -> a further confirmation.
Response #1.24: We corrected the new line 311 as suggested.
Comment #1.25: Discussion, line 304: "R1was..." -> R1 was...
Response #1.25: We added the missing space.
Reviewer 2 Report
Comments and Suggestions for Authors
The present study explores the impact of margin status on long-term outcomes after pancreaticoduodenectomies for PDAC, showing the crucial role of obtaining negative resection margins. This is probably the surgeon's most significant contribution to survival in PDAC, apart from reduced postoperative mortality. The topic is not new; few previous studies have assessed the same issue and reached the same conclusion, but the results of the current study may add value to the current literature. The paper is scientifically sound, well-designed, and well-written; the methods and analyses are correctly used/ performed, and the results sustain the conclusions. The study's main limitation is the relatively low number of analyzed patients during a relatively long time and the fact that it does not bring any novelty to the field.
A few issues should be addressed before acceptance:
Why did the authors exclude neoadjuvant therapies from the analyses? It might be an exciting topic to explore if neoadjuvant therapy is associated with increased rates of negative resection margins, as one might expect.
The missing data about the type and completion of adjuvant therapy is a study limitation that might impact the long-term outcomes. Please state it in the study limitations.
Is it possible to assess the impact of different/ multiple sites of R1 resections on the long-term outcomes in the present series?
No data about associated vascular resection or severe complications are provided; these aspects might also influence the results.
Comments on the Quality of English LanguageMinor editing of English language required
Author Response
GENERAL CONSIDERATIONS
Primarily we would like to thank the Editor and the Reviewers for taking the time to thoughtfully consider our manuscript, and for providing precious observations and suggestions, which have been very helpful to amend the manuscript, as we hope it emerges from the examination of the present document.
It is our feeling that along the line of reviewing the paper we also managed to improve it both in presentation and readability. Please find below our point-by-point response.
Reviewer #2
Comments to the Authors
The present study explores the impact of margin status on long-term outcomes after pancreaticoduodenectomies for PDAC, showing the crucial role of obtaining negative resection margins. This is probably the surgeon's most significant contribution to survival in PDAC, apart from reduced postoperative mortality. The topic is not new; few previous studies have assessed the same issue and reached the same conclusion, but the results of the current study may add value to the current literature. The paper is scientifically sound, well-designed, and well-written; the methods and analyses are correctly used/ performed, and the results sustain the conclusions. The study's main limitation is the relatively low number of analyzed patients during a relatively long time and the fact that it does not bring any novelty to the field.
A few issues should be addressed before acceptance:
Comment #2.1: Why did the authors exclude neoadjuvant therapies from the analyses? It might be an exciting topic to explore if neoadjuvant therapy is associated with increased rates of negative resection margins, as one might expect.
Answer #2.1: We do thank the reviewer for this comment. For instance, two main reasons brought us to exclude patients who underwent NAT. First, the low number of patients (39 out of 221) that would make the results of the statistical comparison questionable. Second, the therapeutic schemes highly varied over the years during the study time elapse. This would have constituted an additional bias for the generalization of the results. However, we are currently working on a multicenter protocol that is aimed to increase the study population and that will take into account the impact of NAT on resection margin status, even in relation to the different regimens employed.
Comment #2.2: The missing data about the type and completion of adjuvant therapy is a study limitation that might impact the long-term outcomes. Please state it in the study limitations.
Answer #2.2: We do completely agree with the reviewer and we do apologize not to have sufficiently stressed this limitation. As suggested, please find the following statement added to the Discussion section (pag. 9, lines 322-324) among the study limitations:
“Finally, data on adjuvant treatment were not available. This inevitably limited the evaluation of the impact of type and duration of the post-operative treatment on long-term outcomes.”
Comment #2.3: Is it possible to assess the impact of different/ multiple sites of R1 resections on the long-term outcomes in the present series?
Answer #2.3: This is an interesting topic, especially in relation to the current evidence in the literature. However, as reported in the Supplementary Table 1, the number of patients per type of positive resection margin is too low to permit us to perform a reliable analysis.
Comment #2.4: No data about associated vascular resection or severe complications are provided; these aspects might also influence the results.
Answer #2.4: We do thank the reviewer for this appropriate comment. Indeed, in our center, the preoperative detection at the CT scan of borderline resectable tumors is always an indication to neoadjuvant treatment, that we defined as an exclusion criterion in our study. However, at the time of surgery, 9 patients needed a tangential vascular resection (superior mesenteric vein/portal vein in all cases) for the intraoperative suspicion of vascular involvement. No difference was noted in terms of number of vascular resections between the R0 and R1 cohorts (p= 0.2). This information was added to Table 1. Moreover, the univariate analysis did not evidence any potential influence of vascular resection on both OS and DFS (p=0.64 and p=0.4, resepectively – please see Table 3).
In addition, please find the following statement added to the Results section (pag. 5, lines 184-186):
“A tangential venous resection was needed in 9 cases (5.4%) for an intraoperative suspicion of vascular invasion, with no difference between the R0 and R1 groups (p=0.2).”
With regards to the influencing role of major complications on long-term outcomes, we do agree with the reviewer on the need to include them for the analysis of OS and DFS, especially in the light of a consistent number of authors that demonstrated a worse prognosis in case of major complications onset. In our study cohort, 35 patients (21%) developed a major complication (defined as a Clavien-Dindo grade³3), with no difference between the two study cohorts (please see the data added to Table 1). However, at the univariate analysis, the occurrence of major complications did not affect OS and DFS, although the p value approached the significance (p=0.07) for DFS. Please, see these data added to Table 3.
Reviewer 3 Report
Comments and Suggestions for Authors
thank you for allowing me to review this retrospective monocentric study whose aim was to evaluate the carcinological impact of R1 resection after cephalic duodenopancreatectomy for ductal adenocarcinoma of the pancreas.
The manuscript is well written, with the primary and secondary objectives clearly expressed.
the analysis of the surgical specimen by inking and the definition of the R1 resection are in line with recommendations in the literature.
It would be useful to specify whether the histological analysis of the surgical specimen was carried out by the same operator, as this may influence the quality of the analysis and the oncological results.
It would also be useful to enclose a flow chart of all patients operated on during the study period, including excluded patients.
In terms of results, an R1 resection was observed in 37% of the cohort. Did the authors observe any variation according to the study period?
90% of R1 patients received adjuvant treatment. What were the reasons for the remaining 10%?
in this study, it is necessary to have morbidity and mortality results. in fact, the occurrence of post-operative complications has been incriminated in the occurrence of local recurrence and in the decrease in recurrence-free survival. why didn't the authors analyze the influence of post-operative complications?
the authors observed no difference in terms of time to recurrence between the two groups. how can this result be explained?
Author Response
GENERAL CONSIDERATIONS
Primarily we would like to thank the Editor and the Reviewers for taking the time to thoughtfully consider our manuscript, and for providing precious observations and suggestions, which have been very helpful to amend the manuscript, as we hope it emerges from the examination of the present document.
It is our feeling that along the line of reviewing the paper we also managed to improve it both in presentation and readability. Please find below our point-by-point response.
Reviewer #3
Comments to the Author
Thank you for allowing me to review this retrospective monocentric study whose aim was to evaluate the carcinological impact of R1 resection after cephalic duodenopancreatectomy for ductal adenocarcinoma of the pancreas.
The manuscript is well written, with the primary and secondary objectives clearly expressed.
The analysis of the surgical specimen by inking and the definition of the R1 resection are in line with recommendations in the literature.
Comment #3.1: It would be useful to specify whether the histological analysis of the surgical specimen was carried out by the same operator, as this may influence the quality of the analysis and the oncological results.
Answer #3.1: We do appreciate this valuable comment. Indeed, as reported in the Methods section (pag. 3, lines 140-143) “all the histopathological slides were retrospectively reviewed by a dedicated pathologist in order to re-assess the R status…” (dr. Giulia Scaglione in the author list). This was an essential step in order to avoid a potential bias in the histopathological quality assessment derived from different operators.
Comment #3.2: It would also be useful to enclose a flow chart of all patients operated on during the study period, including excluded patients.
Answer #3.2: We do thank the reviewer for this comment that permits us to better present the study population in relation to the inclusion and exclusion criteria. Please find the flow chart added as Figure 1 in the Results section (pag. 4), as follows:
Figure 1. Study population flowchart
Thus, the number of the subsequent figures were changed.
Comment #3.3: In terms of results, an R1 resection was observed in 37% of the cohort. Did the authors observe any variation according to the study period?
Answer #3.3: We do appreciate this valuable comment. For instance, we did not evidence any significant difference in terms of R1 rate over the years of the study period. This was probably due to the use of the same surgical technique performed by the same surgeons over the study time elapse.
Comment #3.4: 90% of R1 patients received adjuvant treatment. What were the reasons for the remaining 10%?
Answer #3.4: We do thank the reviewer for this comment. As reported, the 9.7% of the R1 population (6 patients) did not undergo adjuvant treatment. Specifically, 4 patients refused the post-operative treatment while 2 patients were excluded for poor performance status. Given the importance of such an information, the following statement was added to the Result section (pag. 5, lines 192-194):
“In this last regard, 6 R1 patients (9.7%) did not undergo adjuvant treatment due to patient refusal (4 cases) and poor post-operative performance status (2 cases)”
Comment #3.5: In this study, it is necessary to have morbidity and mortality results. in fact, the occurrence of post-operative complications has been incriminated in the occurrence of local recurrence and in the decrease in recurrence-free survival. why didn't the authors analyze the influence of post-operative complications?
Answer #3.5: We do appreciate this observation and we do agree with the reviewer on the need to investigate the prognostic role of major complications on OS and DFS, especially in the light of a consistent number of authors that demonstrated a worse prognosis in case of major complications onset. In our study cohort, 35 patients (21%) developed a major complication (defined as a Clavien-Dindo grade³3), with no difference between the two study cohorts (please see the data added to Table 1). However, at the univariate analysis, the occurrence of major complications did not affect OS and DFS, although the p value approached the significance (p=0.07) for DFS. Please, see these data added to Table 3.
With regards to mortality, 6 patients (2.7%) died post-operatively (please see Figure 1)
Comment #3.6: The authors observed no difference in terms of time to recurrence between the two groups. how can this result be explained?
Answer #3.6: We do thank the reviewer for the comment. According to our findings, R1 patients presented a significantly higher rate of tumor recurrence but similar time to recurrence in comparison to the R0 cohort. This last data may find justification in the high recurrence rate in the whole population (68.3%) for a biologically aggressive disease that implies poor long-term outcomes per se, independently of the resection margin status. In addition to this, it is likely that the small sample size of the study cohorts has brought to similar time to recurrence values.
Reviewer 4 Report
Comments and Suggestions for Authors
Quero et al. evaluated the role of resection margin status and survival after pancreatectomy for ductal adenocarcinnoma of the pancreas. In 167 patients R0 resection was achieved in 62.8% of cases. R1 was associated with a decreased OS, while R! anf lymph node involvement were only independent prognostic factors for OS.
Free resection margins is a well-recognized prognostic factor for PDAC but, as Authors stated, this parameter has not been universally accepted. So, this paper confirm previous reports about the importance of R0 in predicting survival of patients who underwent pancreatic resection.
I have some issues from this manuscript:
1- The main problem is the retrospective nature of the study. So, the different margin involved by the tumor is not specified and informations about eventual treatment (i.e. radiotherapy) and its impact on patients prognosis is not reported.
2-There are many factors potentially able to influence the margin status after resection. It is not clear the proportion of resectable vs border-line resectable or locally advanced PDAC and the eventual diffrence in R1 frequency. Moreover, the rate of vascular resection is not reported.
3-The Authors in the section Discussion, lines 275-278, and in section Conclusion, lines 323-324, emphasize the potential benefit of NAT also in resectable PDAC, both from their own data and from previous publications reporting R1 reduction rate and improvement in long-term outcome. However, there are contrasting reports about the usefulness of NAT in resectable PDAC (DOI: 10.1016/S2468-1253(23)00405-3) and the results of this study does not support this conclusion.
4- Data about the adjuvant therapy in this series are lacking. Since neoadjuvant therapy(NAT) is increasingly used in clinical practice, it could be interesting to know how many patients underwent NAT in the Authors' Institution.
Author Response
GENERAL CONSIDERATIONS
Primarily we would like to thank the Editor and the Reviewers for taking the time to thoughtfully consider our manuscript, and for providing precious observations and suggestions, which have been very helpful to amend the manuscript, as we hope it emerges from the examination of the present document.
It is our feeling that along the line of reviewing the paper we also managed to improve it both in presentation and readability. Please find below our point-by-point response.
Reviewer #4
Quero et al. evaluated the role of resection margin status and survival after pancreatectomy for ductal adenocarcinnoma of the pancreas. In 167 patients R0 resection was achieved in 62.8% of cases. R1 was associated with a decreased OS, while R! anf lymph node involvement were only independent prognostic factors for OS.
Free resection margins is a well-recognized prognostic factor for PDAC but, as Authors stated, this parameter has not been universally accepted. So, this paper confirm previous reports about the importance of R0 in predicting survival of patients who underwent pancreatic resection.
I have some issues from this manuscript:
Comment #4.1: The main problem is the retrospective nature of the study. So, the different margin involved by the tumor is not specified and informations about eventual treatment (i.e. radiotherapy) and its impact on patients prognosis is not reported.
Answer #4.1: We do thank the reviewer for this comment, and we do agree with this observation. For instance, the retrospective study design and the lack of evaluation of the long-term outcomes in relation to the type of involved resection margin represent a limitation. This has been already reported as a drawback of the study at the end of the Discussion section (pag. 9, lines 315-319). In addition, we added the following statement in order to highlight the absence of information on the adjuvant treatment as a further limitation (pag. 9, lines 322-324):
“Finally, data on adjuvant treatment were not available. This inevitably limited the evaluation of the impact of type and duration of the post-operative treatment on long-term outcomes.”
However, we are currently working on a multicenter protocol in order to increase the sample size of the study population and to specifically evaluate the influencing role of the involved resection margin and adjuvant treatment.
Comment #4.2: There are many factors potentially able to influence the margin status after resection. It is not clear the proportion of resectable vs border-line resectable or locally advanced PDAC and the eventual diffrence in R1 frequency. Moreover, the rate of vascular resection is not reported.
Answer #4.2: We do appreciate this comment. Indeed, in our center, the preoperative detection at the CT scan of borderline and locally advanced tumors was always an indication to neoadjuvant treatment, that we defined as an exclusion criterion. However, at the time of surgery, 9 patients needed a tangential vascular resection (superior mesenteric vein/portal vein in all cases) for the intraoperative suspicion of vascular involvement. No difference was noted in terms of number of vascular resections between the R0 and R1 cohorts (p= 0.2). This information was added to Table 1. Moreover, the univariate analysis did not evidence any potential influence of vascular resection on both OS and DFS (p=0.64 and p=0.4, resepectively – please see Table 3).
Please find the following statement added to the Results section (pag. 5, lines 184-186):
“A tangential venous resection was needed in 9 cases (5.4%) for an intraoperative suspicion of vascular invasion, with no difference between the R0 and R1 groups (p=0.2).”
Comment #4.3: The Authors in the section Discussion, lines 275-278, and in section Conclusion, lines 323-324, emphasize the potential benefit of NAT also in resectable PDAC, both from their own data and from previous publications reporting R1 reduction rate and improvement in long-term outcome. However, there are contrasting reports about the usefulness of NAT in resectable PDAC (DOI: 10.1016/S2468-1253(23)00405-3) and the results of this study does not support this conclusion.
Answer #4.3: We do thank the reviewer for this interesting comment, and we do apologize not to have taken into consideration the suggested article. With regards to our study cohort, as reported in the Methods section (pag. 2; lines 90-91), NAT was considered an exclusion criterion for our analysis. Thus, no conclusion was drawn on NAT influence on long-term outcomes based on our study cohort.
On the counterpart, regarding the evidence present in the literature on this topic, please find the following modification according to the appropriate reviewer’s suggestion:
Discussion section (pag. 9, lines 284-286)
“Indeed, the impressive survival advantages of FOLFIRINOX chemotherapy are increasingly emphasized, and the clear benefits in terms of tumor downstaging and improvement in resection rates in borderline lesions are currently paving the way to its application to even resectable PDACs, although with currently reported promising contrasting results are currently present in the literature on this last topic. in terms of R1 reduction rate and amelioration of the long-term outcomes [42-44].”
Moreover, the statement “Moreover, given the evident benefits of the neoadjuvant treatment in reducing tumor size, making PD less challenging for the surgeon and aiding the achievement of tu-mor-free margins, its potential application to resectable lesion may potentially give a further contribution in reducing the R1 rate.” in the Conclusion section (pag. 10, lines 331-334) was removed.
In addition, we added the suggested reference as reference 44.
Comment #4.4: Data about the adjuvant therapy in this series are lacking. Since neoadjuvant therapy(NAT) is increasingly used in clinical practice, it could be interesting to know how many patients underwent NAT in the Authors' Institution.
Answer #4.4: For instance, the lack of information on the type and frequency of adjuvant treatment represents a limitation of our study. To better outline this drawback, the following statement was added to the limitations of the study (pag. 9, lines 322-324):
“Finally, data on adjuvant treatment were not available. This inevitably limited the evaluation of the impact of type and duration of the post-operative treatment on long-term outcomes.”
With regards to NAT, this was considered an exclusion criterion for our study purposes, as reported in the Methods section (pag. 2; lines 90-91) and in the added Figure 1. However, as a whole, during the study period, 39 patients out 221 PDs underwent NAT (Figure 1) but excluded from analysis.
Round 2
Reviewer 2 Report
Comments and Suggestions for Authors
The authors correctly addressed all major concerns raised by the reviewers
Reviewer 3 Report
Comments and Suggestions for Authors
the authors have responded point by point to questions and comments that have significantly improved the value of the manuscript.
Reviewer 4 Report
Comments and Suggestions for Authors
The Authors adequately replied to my suggestions. The paper is suitable for publication.